# Associations of nuptiality perceptions, financial difficulties, and socio-demographic factors with mental health status in Australian adults: Analysis of the Household, Income and Labour Dynamics in Australia (HILDA) survey

**Bernard Kwadwo Yeboah Asiamah-Asare**[1,2], **Prince Peprah**[3,4], **Collins Adu**[5,6]*, **Bright Opoku Ahinkorah**[7,8], **Isaac Yeboah Addo**[6,9]

1 Institute of Applied Health Sciences, University of Aberdeen, Aberdeen, Scotland, 2 Curtin School of Population Health, Curtin University, Perth, Western Australia, Australia, 3 Social Policy Research Centre, UNSW, Sydney, Australia, 4 Centre for Primary Health Care and Equity, UNSW, Sydney, Australia, 5 Department of Health Promotion, Education and Disability Studies, School of Public Health, Kwame Nkrumah University of Science and Technology, Kumasi, Ghana, 6 Centre for Social Research in Health; UNSW Sydney, Sydney, Australia, 7 School of Public Health, University of Technology Sydney, Sydney, Australia, 8 School of Clinical Medicine, UNSW Sydney, Sydney, Australia, 9 Concord Clinical School, University of Sydney, Sydney, Australia

* collinsadu80@yahoo.com

## Abstract

### Objective

This study examined the association of nuptial/relationship factors, financial difficulties, and socio-demographic factors with the mental health status of Australian adults.

### Design

Cross-sectional quantitative study design.

### Settings, participants, and interventions

Using data from the Household, Income and Labour Dynamics in Australia (HILDA) survey wave 19, 6846 adults were included in the analysis. Mental health was measured using the mental component summary (MCS) subscale of the Short-Form Health Survey SF-36. Hierarchical multiple linear regressions were used to examine the predictors of mental health status.

### Results

Overall, 7.1% of the participants reported poor mental health status. Individual financial difficulty factors explained 3.2% (p<0.001) of the variance in mental health scores. In addition, financial difficulties were negatively associated with mental health status. Nuptiality and relationship factors accounted for 9.8% (p<0.001) of the variance in mental health status.

**Data Availability Statement:** The data underlying this study are derived from the Household, Income and Labour Dynamics in Australia (HILDA) Survey. These data contain potentially identifying or sensitive patient information and are owned by a third-party organization. Requests for data access and associated forms can be submitted via email to the Department of Social Services (DSS) at LongitudinalStudiesDataAccess@dss.gov.au and to the Australian Data Archive (ADA) at ada@ada.edu.au. The data can be accessed through the following link: https://dataverse.ada.edu.au/dataverse.xhtml?alias=hilda.

**Funding:** The authors received no specific funding for this work.

**Competing interests:** The authors have declared that no competing interests exist.

## Conclusion

The study suggests negative marital or relationship perceptions and financial difficulties are significant factors accounting for poor mental health. This finding suggests the need for more policy attention toward the social determinants of poor mental health especially nuptiality or relationship perceptions which have received less policy and research attention in Australia.

## Introduction

Mental health constitutes an important aspect of a person's life [1]. It can be defined as "a state of well-being in which individuals realise their own abilities, can cope with the normal stresses of life, can work productively and fruitfully, and are able to make a contribution to their community" [2]. Evidence suggests that poor mental health is associated with past and ongoing adverse issues in life, such as employment difficulties, financial hardships, and domestic violence [3, 4]. Research also shows that people with mental health issues have a two-to-threefold higher risk of mortality when admitted to a hospital compared to those with no mental health issues [5, 6].

Recent studies have increasingly focused on the social determinants of poor mental health with evidence showing a variety of findings across different countries and social contexts [4, 7]. Kiely et al. [4] have established in a narrative review that women generally have poorer mental health than men in terms of depression and anxiety symptoms, but the gender differences decline with age. Another systematic review and meta-analysis has also indicated that individuals who attempted self-immolation were mainly women, married, and young adults [8]. Again, a meta-analysis of longitudinal studies has further shown that shift work is associated with an increased risk of adverse mental health issues, particularly, depressive symptoms [9]. An emerging trend in these recent studies is that mental health affects and is affected by multiple socio-economic factors that may vary across countries and jurisdictions.

In Australia, mental health conditions are common problems with statistics showing that one in five (20%) Australians had a mental or behavioural condition in the 2017–2018 financial year [10]. Preliminary results from the Australian Bureau of Statistics have further indicated that at least 15% of the population aged 16–85 years experienced high or very high levels of psychological distress in 2021 [11]. These statistics suggest a high risk of mental health conditions in the country, indicating the need for progressive research into the determinants of these risks.

Several studies in the country have examined associations of poor mental health with housing affordability [12], sleep patterns [13], employment [14], personality [15], community participation [16], physical activity [17], urbanicity [18], age and disability [19], recurring pain [20], and migration [21]. However, little is known about the interplay of nuptiality or relationship factors, financial difficulties, socio-demographic factors, and mental health in Australia. Nuptiality in the context of our study refers to the state of being married, the frequency of being married, characteristics of marriages, marital experiences, and dissolution of marriages in a population [22]. We hypothesised that nuptiality factors, such as number of times legally married, personal rating of marriage experience, marital wishes, expectations of marriage, and problems in relationships will have a significant association with mental health status. In this paper, we examined the associations between these variables using data from the Household,

Income and Labour Dynamics in Australia (HILDA) survey. Findings in this study are important for developing more tailored public health policies on mental health in the country and might be useful for health planners, mental health advocates, social workers, and the Australian government.

## Methods

### Data description and sample

Data for this study were obtained from the Household, Income and Labour Dynamics in Australia (HILDA) survey, which commenced in 2001. Wealth, labour market outcomes, household and family ties, fertility, health, and education are all included in the dataset. To pick an initial sample, a multistage sampling strategy was adopted. To begin, 488 Census Collection Districts are selected using a probability proportionate to size sampling approach (CDs). Each of the districts has between 200 and 250 households. Second, from each of the CDs, a random sample of 22–34 residences was chosen. Finally, a maximum of three households were chosen from each dwelling, totaling 12,252 households. Since 2001, the annual data collection has included a sample of household members aged 15 and up. Certified enumerators conducted face-to-face and telephone interviews to collect data. In this case, a self-completed questionnaire was employed in accordance with the University of Melbourne's ethical guidelines. Over time, the sample size was increased. It comprises any child born or adopted by a group of respondents, as well as any new household member resulting from changes in the source families' composition. As a result, the poll has a total annual coverage of nearly 17,000 Australian people. The sampling technique, study design, and data collection strategies for the waves have all been discussed in depth elsewhere [23]. There are currently 19 waves of the HILDA survey however, this study utilised the most recent wave (wave 19) as our variables of interest were in this specific wave. Like all other waves, wave 19 contains detailed information on participants' socio-demographics and issues on mental health. To avoid potential bias, missing observations on the outcome variable (mental health) were excluded. The final analytic sample consists of 6,846 observations.

### Measures

**Mental health.** Mental health was measured using the Short-Form Health Survey SF-36 [24, 25]. The SF-36 measures quality of life in the last 4 weeks, with a mental component summary (MCS) subscale. Five items from the MCS subscale (including been a nervous person, felt so down in the dumps nothing could cheer you up, felt calm and peaceful, felt down, and been a happy person) and scored on a 5-point Likert scale (0 = none of the time to 5 = all of the time) (Cronbach's alpha: $\alpha$ = 0.83) were used to assess mental health. The raw scores were transformed, and a composite variable was created by adding the scores for each participant to generate a total score of between 0 and 100 in line with the SF-36 scale scoring instructions. A higher score was suggestive of better mental quality of life and a score of less than 50 is taken as having a poor mental health status [24, 25].

**Financial difficulty variable.** History of financial difficulties was assessed on several variables including "Difficulty paying utility bills on time", Difficulty paying mortgage or rent", "Pawned or sold something", "Went without meals", "Was unable to heat home", Financial help from friends/family" and "Help from welfare/com organisations". Participants were asked to respond "Yes" or "No" to the list of items. Here, no composite variable was generated. All the individual items measured were included in the analysis.

**Nuptiality and relationships perceptions.** Nuptiality was assessed using participants' number of times of legal marriage. This was measured using the item: "How many times have

you been legally married?" scored on the scale: "Once, twice, three-time, four times". Participants' relationship perceptions were assessed using six items including: "How good is your relationship compared to most" rated on a scale: 1 = poor to 5 = excellent; "How often do you wish you had not been married or got in the relationship" rated on the scale: 1 = never to 5 = very often; "To what extent has your relationship met original expectations" rated on the scale: 1 = hardly at all to 5 = completely; "How much do you love spouse/partner" rated on scale 1 = not much to 5 = very, very much; "How many problems are there in your relationship" rated on the scale: 1 = not much to very much; and "How well do your spouse/partner meet your needs" rated on a Likert scale: 1 = poor to 5 = excellent.

**Analysis plan.** Data were analysed using STATA version 13 (StataCorp LP, College Station, TX). Descriptive statistics were assessed for continuous and categorical variables. Using one-way analysis of variance (ANOVA), the differences in mental health status across the different socio-demographic characteristics of participants were examined. Hierarchical multiple linear regression was employed in examining the relative contribution of socio-demographic characteristics, financial difficulties, nuptiality and relationship perceptions to mental health. Three models were hierarchically specified; socio-demographic factors were first entered in model 1, financial difficulties variables were then introduced in model 2 and nuptiality and relationships perceptions factors were entered at the final stage in model 3. The socio-demographic factors were entered first in step 1 to allow for the distinctive effects of financial difficulties and nuptiality and relationship perceptions factors to be ascertained. The level of statistical significance was set at $p < 0.05$ at both the bivariate and multiple variable level analyses.

**Ethics and data availability.** The HILDA dataset is publicly available and accessible by authorised researchers and data users who have obtained permission from the DSS. PP completed a data access form, which was approved, and signed a deed of license. All participants of HILDA provided written informed consent following explanation of the study.

## Results

### Background characteristics of study participants

A total of 6846 study participants were included in the analysis. Majority of the participants were aged above 42 years (60.9%), females (51.4%), born in Australia (77.5%), and married (78.2%). More of the participants had year 11 certificate and below (27.7%) and 70% of them were employed. The mean MCS scores among study participants was 76.4±15.8, (range 4–100). Participants found to have poor mental health status (MCS score less than 50) were 7.1%. Table 1 presents the background characteristics of the study participants.

### Factors associated with mental health status

The multiple linear regression results are shown in Table 2. The background characteristics accounted for 2.1% of the variance in MCS scores. The study participants aged 60 years and above compared to those aged less than 25 years (β = 3.29, 95%CI = 1.21–5.37) were associated with higher scores of mental health whereas being a female (β = -1.14, 95%CI = -1.85- -0.43), born outside of Australia (β = -1.60, 95%CI = -2.44- -0.75), retired (β = -2.99, 95%CI = -4.34- -1.65) and being a student (β = -4.97, 95%CI = -7.49- -2.46) were associated with lower scores of mental health.

Individual history of financial difficulties accounted for an additional 3.2% of the variance on MCS scores, with participants who had difficulties paying utility bills on time (β = -2.05, 95%CI = -3.49- -2.46), pawned or sold some belongings (β = -3.60, 95%CI = -5.81- -1.39), and

Table 1. Background characteristics of study participants by mental health scores (N = 6846).

| Characteristics | Total, n (%) | Mental health scores | p-value |
|---|---|---|---|
| **Age in years** | | | <0.001 |
| 16–24 | 374(5.5) | 74.2±16.2 | |
| 25–33 | 995(14.5) | 75.9±14.3 | |
| 34–42 | 1308(19.1) | 76.1±15.7 | |
| 43–51 | 1446(21.1) | 75.8±15.8 | |
| 52–60 | 1142(16.7) | 76.4±16.0 | |
| 60+ | 1581(23.1) | 77.8±16.4 | |
| **Sex** | | | <0.001 |
| Male | 3327(48.6) | 77.3±15.3 | |
| Female | 3519(51.4) | 75.5±16.2 | |
| **Place of birth** | | | 0.006 |
| Australia | 5303(77.5) | 76.6±15.6 | |
| Other | 1543(22.5) | 75.4±16.3 | |
| **Education level** | | | <0.001 |
| Postgraduate degree | 812(11.9) | 77.4±14.3 | |
| Bachelor's degree | 1066(15.6) | 77.5±13.9 | |
| Diploma qualification | 706(10.3) | 77.1±14.8 | |
| Cert III or IV | 1533(22.4) | 76.6±15.7 | |
| Year 12 certificate | 835(12.2) | 76.1±16.7 | |
| Year 11 certificate and below | 1894(27.7) | 74.9±17.3 | |
| **Marital status** | | | 0.001 |
| Married | 5350(78.2) | 76.7±15.6 | |
| Separated/Divorced/Widowed | 482(7.0) | 75.6±17.5 | |
| De-facto/co-habiting | 1014(14.8) | 74.8±15.8 | |
| **Employment status** | | | <0.001 |
| Employed | 4791(70.0) | 76.9±14.9 | |
| Not employed | 129(1.9) | 72.8±18.0 | |
| Retired | 1278(18.7) | 76.5±17.1 | |
| Home duties | 512(7.5) | 73.6±18.1 | |
| Student | 34(0.5) | 71.3±15.1 | |
| Other | 102(1.5) | 68.9±22.0 | |

sought financial help from friends/family (β = -3.20(-4.60- -1.80) or sought help from welfare/ community organisations (β = -2.61, 95%CI = -5.05- -0.17) scored low on mental health.

Nuptial and relationship characteristics accounted for 9.8% of the variance on MCS scores. There were significant positive associations between mental health scores and the extent to which participants perceived their relationships as good (β = 2.22, 95%CI = 1.51–2.92) and meet their original expectations (β = 1.37, 95%CI = 0.78–1.96), indicating that participants who highly perceived their relationships as good and meeting their original expectations had better mental health status. However, there were significant negative associations between mental health scores and the extent to which participants wish not to be married or be in a relationship (β = -0.62, 95%CI = -1.18- -0.07), love spouse/partner (β-1.52, 95%CI = -2.24- -0.81) and the frequency of problems in the relationship (β = -2.63, 95%CI = -3.05- -2.21). This suggests participants who very often wished not to have been married or got into the relationship, very much loved their spouses or partners and experienced many problems in their relationships were less likely to report better mental health status.

**Table 2. Multiple linear regression of predictors of mental health among study participants.**

| Characteristics | Mental health scores | | |
|---|---|---|---|
| | Model 1 β(95%CI) | Model 2 β(95%CI) | Model 3 β(95%CI) |
| | Adjusted R$^2$ = 0.0208*** | Adjusted R$^2$ = 0.0525*** | Adjusted R$^2$ = 0.1502*** |
| **Age in years** | | | |
| 16–24 | Reference category | Reference category | Reference category |
| 25–33 | 0.54(-1.37–2.46) | 0.20(-1.69–2.10) | 0.03(-1/77-1.82) |
| 34–42 | 0.64(-1.28–2.56) | 0.09(-1.81–1.99) | 1.24(-0.57–3.04) |
| 43–51 | 0.57(-1.38–2.53) | -0.38(-2.31–1.56) | 1.27(-0.57–3.12) |
| 52–60 | 1.65(-0.37–3.66) | 0.24(-1.75–2.24) | 0.99(-0.91–2.89) |
| 60+ | 5.12(2.91–7.32)*** | 3.42(1.23–5.61)** | 3.29(1.21–5.37)** |
| **Sex** | | | |
| Male | Reference category | Reference category | Reference category |
| Female | -1.67(-2.43- -0.92)*** | -1.69(-2.43- -0.95)*** | |
| **Place of birth** | | | |
| Australia | Reference category | Reference category | Reference category |
| Other | -1.79(-2.70- -0.88)*** | -1.77(-2.67- -0.88)*** | -1.60(-2.44- -0.75)*** |
| **Education level** | | | |
| Postgraduate degree | Reference category | Reference category | Reference category |
| Bachelor degree | 0.40(-1.04- -1.83) | 0.41(-1.01–1.82) | 0.51(-0.83–1.85) |
| Diploma qualification | -0.44(-2.02- -1.14) | -0.04(-1.60–1.51) | 0.28(-1.20–1.75) |
| Cert III or IV | -0.92(-2.26- -0.43) | -0.24(-1.57–1.09) | 0.50(-0.76–1.76) |
| Year 12 certificate | -0.70(-2.23- -0.83) | -0.09(-1.60–1.43) | 0.04(-1.40–1.47) |
| Year 11 certificate and below | -2.61(-3.93- -1.29)*** | -1.65(-1.65- -0.34)* | -0.94(-2.18–0.30) |
| **Marital status** | | | |
| Married | Reference category | Reference category | Reference category |
| Separated/Divorced/Widowed | -0.98(-2.45- -0.49) | -0.14(-1.59–1.31) | 0.47(-0.91–1.84) |
| De-facto/co-habiting | -1.21(-2.41- -0.01)* | -0.25(-1.44–0.93) | -0.82(-2.29–0.64) |
| **Employment status** | | | |
| Employed | Reference category | Reference category | Reference category |
| Not employed | -3.03(-5.79- -0.26)* | -0.55(-3.29–2.20) | -0.44(-3.04–2.15) |
| Retired | -3.36(-4.80- -1.92)*** | -2.93(-4.34- -1.51)*** | -2.99(-4.34- -1.65)*** |
| Home duties | -3.07(-4.51- -1.63)*** | -1.93(-3.35- -0.50)** | -1.33(-2.68–0.02) |
| Student/Other | -7.18(-9.86- -4.50)*** | -4.91(-7.56- -2.26)*** | -4.97(-7.49- -2.46)*** |
| **Financial difficulty variables** | | | |
| Difficulty paying utility bills on time | | | |
| No | | Reference category | Reference category |
| Yes | | -3.26(-4.75- -1.77)*** | -2.05(-3.49- -2.46)** |
| Difficulty paying mortgage or rent | | | |
| No | | Reference category | Reference category |
| Yes | | -0.97(-2.95–1.01) | -0.37(-2.25–1.51) |
| Pawned or sold something | | | |
| No | | Reference category | Reference category |
| Yes | | -3.56(-5.89- -1.23)** | -3.60(-5.81- -1.39)** |
| Went without meals | | | |
| No | | Reference category | Reference category |
| Yes | | -4.67(-7.66–1.69)** | -2.57(-5.40–0.26) |
| Unable to heat home | | | |
| No | | Reference category | Reference category |

*(Continued)*

**Table 2.** (Continued)

| Characteristics | Mental health scores | | |
|---|---|---|---|
| Yes | | -3.29(-6.55- -0.04)* | -2.42(-5.50–0.67) |
| Financial help from friends/family | | | |
| No | | Reference category | Reference category |
| Yes | | -4.07(-5.54- -2.59)*** | -3.20(-4.60- -1.80)*** |
| Help from welfare/com organisations | | | |
| No | | Reference category | Reference category |
| Yes | | -3.36(-5.93- -0.78)* | -2.61(-5.05- -0.17)* |
| **Nuptiality and relationship variables** | | | |
| Number of times legally married | | | -0.74(-1.57–0.09) |
| Relationship good compared to most | | | 2.22(1.51–2.92)*** |
| Wish not to be married or be in the relationship | | | -0.62(-1.18- -0.07)* |
| Relationship meets original expectations | | | 1.37(0.78–1.96)*** |
| Love spouse/partner | | | -1.52(-2.24- -0.81)*** |
| Problems in relationship | | | -2.63(-3.05- -2.21)*** |
| Spouse/partner meet my needs | | | 0.20(-0.43–0.82) |

*p<0.05, **p<0.01, *p<0.001

## Discussion

Poor mental health is a significant global problem [26]. In Australia, emerging evidence shows that multiple factors may be associated with mental health deterioration, with housing problems, work issues, gender, and age, recognised as key determinants [9, 11, 27–30]. This study builds on previous studies by examining the associations of nuptiality or sexual relationship perceptions, financial difficulties, and socio-demographic factors with mental health in Australia.

Generally, the average mental health status for participants were moderately good but this result should be taken cautiously considering that some participants may have overstated their actual mental health conditions due to the societal stigma associated with mental health issues [31]. Importantly, there were about 7% of participants who reported poor mental health status, and may require mental health support. Our findings further demonstrate that socio-demographic factors, nuptiality or relationship factors, and financial difficulties contributed in various degrees to mental health deterioration in the survey year. In terms of socio-demographic factors, older age ($\geq$ 60 years) was significantly associated with higher mental health scores compared with younger aged ($<$ 25 years). This finding is consistent with preliminary findings by the Australian Bureau of Statistics in 2021, which showed that one in five (20%) Australians aged 16–34 years experienced higher levels of psychological distress compared to those aged 65–85 years (9%) [11]. In the United States, a national survey has similarly revealed that older adults have a lower prevalence of most psychological disorders compared to younger adults [32]. The study also found that participants born overseas, those who self-identified as female, and students had relatively better mental health status compared with those born in Australia, those who identified as males, and non-students, respectively. These findings corroborate previous analyses on mental health trajectories among females [11] and students [33] in Australia. However, we found no existing study that has compared the mental health status of overseas-born residents with Australian-born residents except for one study that suggests that immigrant groups in Australia vary widely in their mental health outcomes [21]. Possible

underlying causes of the poor mental health status among younger age groups, females, overseas-born residents, and students may include structural factors that deepen existing disadvantages for these population groups in Australia and individual-level factors, such as differential coping strengths against stress and other pressures of society [4]. Further exploration into these findings will be useful in developing more tailored interventions to address the mental health needs of the country.

Financial issues are, considerably, one of the commonly cited reasons for poor mental health globally [34]. Two different systematic reviews have shown that economic downturns and their associated moderators, such as unemployment, revenue decline, and debts are significantly associated with poor mental health, including common mental disorders, substance-related conditions, and suicidal behaviours [34, 35]. Consistent with these reviews, findings from this present study showed that participants who were not gainfully employed, had difficulty paying utility bills on time, had difficulty paying mortgage or rent, pawned or sold their assets, went through days without meals, sought financial help from friends or family, and sought help from welfare organisations had poor mental health status. These findings indicate the need to develop structural policies to address financial challenges faced by vulnerable groups in society. Otherwise, persistent stress and burden associated with meeting livelihood needs for disadvantaged groups in Australia, such as those unemployed and without financial stimulus support will only worsen the mental health issues in the country.

Our findings on the associations between nuptiality or sexual relationship perceptions and mental health reveal an interesting paradox. Participants who particularly perceived their relationships as good and meeting their original expectations had better mental health status. Those who wished not to be married or be in their current relationship, those who confirmed to love their partners, and those who mentioned that they were experiencing many problems in their relationships reported poor mental health status. While it is logical to assume that negative perceptions and experiences in marriage or sexual relationships can contribute to mental health, the finding that those who loved their partners also had poor mental health scores was unexpected. Potential confounding factors must be acknowledged in this surprising finding as the implication from the result is that love alone is inadequate for a healthy mental health status in a sexual relationship.

Together, these findings highlight the importance of understanding the impact of nuptiality or sexual relationship factors on mental health. Interestingly, research about the associations of nuptiality or sexual relationship issues with mental health has largely focused on domestic violence [36–38]. These findings can be considered "a wake-up call" for researchers, mental health advocates, and policymakers to consider broadening existing nuptiality-centred research and interventions beyond matters about domestic violence alone.

To this end, we propose early, structural, and co-designed interventions to address nuptiality or sexual relationship factors, financial issues, and socio-demographic factors associated with poor mental health in Australia. One challenge, however, is that addressing these issues will require efforts from multiple sectors of society and it may take time and dedicated efforts to build collaborations among stakeholders, including, the federal government, local politicians, social workers, and health workers [39]. However, developing practical interventions, policies, and securing resources to address these factors associated with poor mental health is feasible and essential. For instance, utility bills could be further subsidised for unemployed people facing consistent challenges in payments. An example of this approach is the wide consultation that was implemented in Kent, England, as part of the process of promoting mental health in the country [39].

## Strengths and limitations

This study has a number of strengths. Firstly, our analysis is based on a national dataset comprising a large sample of participants, which increases the generalisability of the findings of the study. The large sample size enabled us to explore associations of nuptiality or relationship perceptions, financial difficulties, and socio-demographic factors with mental health status using linear regression models. Thus, we were able to specify the most influential variables associated with mental health. The study also expands findings in previous studies focused on social determinants of mental health by including new variables, such as nuptiality or relationship perceptions and financial difficulties. The study also used the longer version of the mental health assessment instrument (SF-36) in measuring mental health which is widely accepted and validated. However, this study acknowledges some limitations. First, participant responses were based on self-reports rather than objective measures of mental health scores, and as a result, the findings may be subjected to social desirability and recall bias. Due to the cross-sectional nature of our study, we could not establish causality, hence the findings should be interpreted cautiously. It is plausible that personality, genetic, and psychosocial variables may be associated with mental health, but we were unable to explore such important variables in this study due to data limitations.

## Conclusion

This study concludes that negative marital or relationship perceptions and financial difficulties are associated with poor mental health status in Australia. Persons who are females, unemployed, and younger are also predisposed to a higher risk of mental health issues. These results suggest the need for more policy attention toward the social determinants of poor mental health especially nuptiality or relationship perceptions, which have received less policy and research attention in Australia. Future studies may consider exploring these factors longitudinally and with qualitative research designs to provide more in-depth interpretations of these results.

## Acknowledgments

This paper uses unit record data from the Household, Income and Labour Dynamics in Australia (HILDA) Survey. The HILDA Project was initiated and is funded by the Australian Government Department of Social Services (DSS) and is managed by the Melbourne Institute of Applied Economic and Social Research (Melbourne Institute). The findings and views reported in this paper, however, are those of the authors and should not be attributed to either the DSS or the Melbourne Institute.

## Author Contributions

**Conceptualization:** Prince Peprah.

**Data curation:** Bernard Kwadwo Yeboah Asiamah-Asare, Prince Peprah.

**Formal analysis:** Prince Peprah.

**Methodology:** Bernard Kwadwo Yeboah Asiamah-Asare, Bright Opoku Ahinkorah.

**Project administration:** Isaac Yeboah Addo.

**Supervision:** Collins Adu.

**Writing – original draft:** Bernard Kwadwo Yeboah Asiamah-Asare, Prince Peprah, Collins Adu, Bright Opoku Ahinkorah, Isaac Yeboah Addo.

**Writing – review & editing:** Bernard Kwadwo Yeboah Asiamah-Asare, Prince Peprah, Collins Adu, Bright Opoku Ahinkorah, Isaac Yeboah Addo.

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
