## [Decision Letter · Decision Letter 0]

28 Feb 2023

PONE-D-23-03512Associations of nuptiality perceptions, financial difficulties, and socio-demographic factors with mental health in Australian adults: Analysis of the Household, Income and Labour Dynamics in Australia (HILDA) surveyPLOS ONE

Dear Dr. Adu,

Thank you for submitting your manuscript to PLOS ONE. After careful consideration, we feel that it has merit but does not fully meet PLOS ONE’s publication criteria as it currently stands. Therefore, we invite you to submit a revised version of the manuscript that addresses the points raised during the review process.

We look forward to receiving your revised manuscript.

Kind regards,

Benojir Ahammed, M.Sc.

Academic Editor

PLOS ONE

Journal Requirements:

Reviewers' comments:

Reviewer's Responses to Questions

**Comments to the Author**

1. Is the manuscript technically sound, and do the data support the conclusions?

Reviewer #1: Yes

Reviewer #2: Yes

2. Has the statistical analysis been performed appropriately and rigorously? 

Reviewer #1: Yes

Reviewer #2: Yes

3. Have the authors made all data underlying the findings in their manuscript fully available?

Reviewer #1: Yes

Reviewer #2: No

4. Is the manuscript presented in an intelligible fashion and written in standard English?

Reviewer #1: Yes

Reviewer #2: Yes

5. Review Comments to the Author

Reviewer #1: This paper is not about mental health status or mental health conditions. MCS is only a measure of mental wellbeing or mental health related quality of life (Mental HRQoL). The literature on mental health conditions are not appropriate for this paper. The terms should be more consistent, and the authors should avoid terms such as status or condition and just use mental wellbeing or mental QOL. They also need to use the new literature that has used MCS not mental health disorders such as depression. MCS is very non-specific. Similarly, there is a need to replicate the findings using logistic regression and test if the results hold with a cut off. Also, we need to see different operationalization of main variable (o vs any) etc.

Is there a dosage effect of your variable?

Reviewer #2: It was a pleasure reading and learning about your work. A very relevant topic that benefits research, policy and practice. Overall your study was succinct, logically fluid and comprehensible. I however encourage listing redundancies and grammato-syntactical errors to enhance readability. Find below my specific comments for considerations.

Abstract

Your abstract is not exhaustive to pass across key details of the study

Avoid using abbreviations you haven’t introduced (e.g. MCS)

Although the HILDA survey was a cross-sectonal study, I think your study was a longitudinal analysis (retrospective) using data drawn from the HILDA survey and not technically a CS study in that sense.

Introduction

When you cite quoted references, include the source and page number within the in-text citation. That is standard. For e.g.,

“a state of well-being in which individuals realise their own abilities, can cope with the normal stresses of life, can work productively and fruitfully, and are able to make a contribution to their community”

Methods

Any ethical considerations for the use of HILDA? Declare it in your methods.

“To begin, 488 Census Collection Districts are selected using a probability proportionate to size sampling approach (CDs). Each of the districts has between 200 and 250 households.”

Take out the antecedent “to begin” and use more engaging word syntax e.g., First, firstly, etc.

Were samples weighted in the iteration your used. If so stated that

Your definitions of measures need more clarification in your SAP. What was your dependent/outcome variable and your predictor variables. They should be clearly defined under measures

Results

I think background characteristics would read better as descriptive characteristics

Discussion

I will suggest a debrief of study objectives with a summary of key findings before diving into discussion. It is more engaging that way.

“Together, these findings highlight the importance of understanding the impact of nuptiality or sexual relationship factors on mental health.”

I take umbrage in this statement not because it is not true but you have not shown in your study that there was a gap in understanding the impact of nuptiality on mental health. Your lit review missed that mark to inform this as a practical implication in your study. the statement “Interestingly, research about the associations of nuptiality or sexual relationship issues with mental health has largely focused on domestic violence [36-38].” Would be better suited in the introduction to create that hook-gap syntax to justify your study.

“we propose early, structural, and co-designed interventions to address nuptiality or sexual relationship factors, financial issues, and socio-demographic factors associated with poor mental health in Australia.”

While this is valid, it is too vague. What are the strategies currently in place in Australia addressing mental issues and nuptiality or domestic violence? what are the gaps in programs, initiatives, and policies in place to address this challenge? How can your study inform these gaps. Give direct, practical, recommendations, based on evidence based practice or policies from other jurisdictions.

References

A mix of superscripts and in-text numerical citations were used. Ensure your citations align with author guidelines

e.g.

“The sampling technique, study design, and data collection strategies for the waves have all been discussed in depth elsewhere. 23”

6. PLOS authors have the option to publish the peer review history of their article (what does this mean?). If published, this will include your full peer review and any attached files.

Reviewer #1: No

Reviewer #2: **Yes: **Udoka Okpalauwaekwe

---

## [Author Response · Author response to Decision Letter 0]

19 Oct 2023

Dear Prof. Benojir Ahammed,

We are grateful to you and the reviewers for your comments on our paper entitled: 

" Associations of nuptiality perceptions, financial difficulties, and socio-demographic factors with mental health in Australian adults: Analysis of the Household, Income and Labour Dynamics in Australia (HILDA) survey". We would also take this opportunity to thank the reviewers for their suggestions. We have taken note of all the comments raised and have responded accordingly as follows. Please be informed that the reviewers' comments are in black whereas our responses are in red. 

Reviewer #1: 

This paper is not about mental health status or mental health conditions. MCS is only a measure of mental wellbeing or mental health related quality of life (Mental HRQoL). The literature on mental health conditions are not appropriate for this paper. The terms should be more consistent, and the authors should avoid terms such as status or condition and just use mental wellbeing or mental QOL.

Response: 

Thank you for this very important observation. We agree that while the MCS score can be used to identify individuals who are experiencing poor mental health-related quality of life, it does not provide a clinical diagnosis of a mental health condition. We have therefore replaced the term “mental health conditions or status” with mental health-related quality of life throughout the paper, including the title. 

 They also need to use the new literature that has used MCS not mental health disorders such as depression. MCS is very non-specific. 

Response: 

Thank you. This section has been revised.

Similarly, there is a need to replicate the findings using logistic regression and test if the results hold with a cut off. 

Response: Thank you for your thoughtful comments. However, we respectfully suggest that such replication is not needed in our study. We utilised a robust hierarchical multiple linear regression and believe that a logistic regression would not necessarily provide additional insights into the relationships between the variables, as the study has already shown significant associations using multiple linear regression. Adding a cut-off point is also not needed, as the study did not classify individuals into discrete categories, but rather used a continuous measure of mental health scores.

Also, we need to see different operationalization of main variable (o vs any) etc.

Is there a dosage effect of your variable?

Response: 

Thank you for your comment. While we appreciate your suggestion, we believe that the current operationalisation of the main variable, mental health, is appropriate for this study. The study utilised a validated measure of mental health, the Mental Component Summary (MCS) score, which is a continuous measure that assesses mental health status based on a range of factors. The study also used a cut-off point of MCS score less than 50 to classify participants with poor mental health status. This approach is commonly used in mental health research and has been shown to be reliable and valid. 

While there may be alternative ways of operationalising mental health, such as using binary outcomes or different cut-off points, we do not believe that such changes would significantly impact the findings of the study. 

Reviewer #2: 

It was a pleasure reading and learning about your work. A very relevant topic that benefits research, policy, and practice. Overall your study was succinct, logically fluid, and comprehensible. I however encourage listing redundancies and grammato-syntactical errors to enhance readability. Find below my specific comments for considerations.

Response: Thank you. All grammato-syntactical errors have checked.

Abstract

Your abstract is not exhaustive to pass across key details of the study

Response: Thank you. We have included more detail in the Abstract as recommended.

Avoid using abbreviations you haven’t introduced (e.g. MCS)

Response: 

Thank you. We have defined the abbreviations at first use.

Although the HILDA survey was a cross-sectonal study, I think your study was a longitudinal analysis (retrospective) using data drawn from the HILDA survey and not technically a CS study in that sense.

Response: 

Thank you. We appreciate your suggestion, however, we respectfully disagree to some extent. Our study did not follow a group of individuals or a cohort over a period of time to examine changes and patterns in behaviour, attitudes, and health outcomes. Rather, we used data collected from a single wave of the survey (wave 19) and conducted a retrospective cross-sectional analysis. Therefore, we opine that our study is not a longitudinal study, but a retrospective cross-sectional study. However, we have made the required correction.

Introduction

When you cite quoted references, include the source and page number within the in-text citation. That is standard. For e.g.,

“a state of well-being in which individuals realise their own abilities, can cope with the normal stresses of life, can work productively and fruitfully, and are able to make a contribution to their community”

Response: Thank you. However, we now do not have quoted references. We have made a significant change to the introduction section based on a critical suggestion by another reviewer.

Methods

Any ethical considerations for the use of HILDA? Declare it in your methods.

Response: 

Thank you. We appreciate your feedback. We have taken your suggestion and revised the "Ethics and data availability" section in our Methods to improve clarity and ensure that the ethical considerations of our study are more prominently displayed.

“To begin, 488 Census Collection Districts are selected using a probability proportionate to size sampling approach (CDs). Each of the districts has between 200 and 250 households.”

Take out the antecedent “to begin” and use more engaging word syntax e.g., First, firstly, etc.

Response: Done. Thank you.

Were samples weighted in the iteration your used. If so stated that

Response: Weighted sample was used and we have indicated that accordingly. Thank you.

Your definitions of measures need more clarification in your SAP. What was your dependent/outcome variable and your predictor variables. They should be clearly defined under measures

Response: Dependent/outcome variable and your predictor variables are been clarified. Thank you.

Results

I think background characteristics would read better as descriptive characteristics

Response: 

Thank you for your suggestion. We appreciate your feedback and agree that using the term "descriptive characteristics" may provide better clarity for the reader. We have revised the manuscript to reflect this change. Thank you again for your valuable feedback.

Discussion

I will suggest a debrief of study objectives with a summary of key findings before diving into discussion. It is more engaging that way.

Response: 

Thank you. This is done.

“Together, these findings highlight the importance of understanding the impact of nuptiality or sexual relationship factors on mental health.”

I take umbrage in this statement not because it is not true but you have not shown in your study that there was a gap in understanding the impact of nuptiality on mental health. Your lit review missed that mark to inform this as a practical implication in your study. the statement “Interestingly, research about the associations of nuptiality or sexual relationship issues with mental health has largely focused on domestic violence [36-38].” Would be better suited in the introduction to create that hook-gap syntax to justify your study.

Response: Thank you. We have moved the statement: “research about the associations of nuptiality or sexual relationship issues with mental health has largely focused on domestic violence [36-38].” to the Introduction section as suggested and deleted all the other statements. 

“we propose early, structural, and co-designed interventions to address nuptiality or sexual relationship factors, financial issues, and socio-demographic factors associated with poor mental health in Australia.”

While this is valid, it is too vague. What are the strategies currently in place in Australia addressing mental issues and nuptiality or domestic violence? what are the gaps in programs, initiatives, and policies in place to address this challenge? How can your study inform these gaps. Give direct, practical, recommendations, based on evidence based practice or policies from other jurisdictions.

Response: Thnak you. We have revised as recommended. However, in our revision, we have taken into consideration the fact that some evidence-based practices and policies from other jurisdictions may not align with our study focus and variables. Therefore, we have taken caution in selecting and recommending interventions that are relevant and applicable to our research context. We believe that our revised paper now provides a more comprehensive and effective approach to addressing the challenges associated with poor mental health. We appreciate your time and consideration

References

A mix of superscripts and in-text numerical citations were used. Ensure your citations align with author guidelines

e.g.

“The sampling technique, study design, and data collection strategies for the waves have all been discussed in depth elsewhere. 23”

Response: Thank you very much. This section has been revised and rectified.

---

## [Decision Letter · Decision Letter 1]

21 Dec 2023

Associations of nuptiality perceptions, financial difficulties, and socio-demographic factors with mental health in Australian adults: Analysis of the Household, Income and Labour Dynamics in Australia (HILDA) survey

PONE-D-23-03512R1

Dear Dr. Adu,

We’re pleased to inform you that your manuscript has been judged scientifically suitable for publication and will be formally accepted for publication once it meets all outstanding technical requirements.

Kind regards,

Benojir Ahammed, M.Sc.

Academic Editor

PLOS ONE

Additional Editor Comments (optional):

Reviewers' comments:

Reviewer's Responses to Questions

**Comments to the Author**

1. If the authors have adequately addressed your comments raised in a previous round of review and you feel that this manuscript is now acceptable for publication, you may indicate that here to bypass the “Comments to the Author” section, enter your conflict of interest statement in the “Confidential to Editor” section, and submit your "Accept" recommendation.

Reviewer #2: All comments have been addressed

2. Is the manuscript technically sound, and do the data support the conclusions?

Reviewer #2: Yes

3. Has the statistical analysis been performed appropriately and rigorously? 

Reviewer #2: Yes

4. Have the authors made all data underlying the findings in their manuscript fully available?

Reviewer #2: Yes

5. Is the manuscript presented in an intelligible fashion and written in standard English?

Reviewer #2: Yes

6. Review Comments to the Author

Reviewer #2: Thank you for allowing me another look at your manuscript. Your thoughtful replies and careful consideration of the reviewers' feedback are commendable. Upon reevaluation, I acknowledge my initial misinterpretation of your study design, having originally mistaken it for longitudinal instead of the correct retrospective cross-sectional approach, and I can see how this has shaped the analysis and conclusions drawn from your data. I wish you the best and look forward to reading more of your work in the future.

7. PLOS authors have the option to publish the peer review history of their article (what does this mean?). If published, this will include your full peer review and any attached files.

Reviewer #2: **Yes: **Udoka Okpalauwaekwe

---

## [Editor Report · Acceptance letter]

16 Jan 2024

PONE-D-23-03512R1 

PLOS ONE

Dear Dr. Adu, 

I'm pleased to inform you that your manuscript has been deemed suitable for publication in PLOS ONE. Congratulations! Your manuscript is now being handed over to our production team.

Kind regards, 

on behalf of

Mr. Benojir Ahammed 

Academic Editor

PLOS ONE